# Antimicrobial Activity of Lavender Essential Oil from *Lavandula angustifolia* Mill.: In Vitro and In Silico Evaluation

**DOI:** 10.3390/antibiotics14070656

**Published:** 2025-06-28

**Authors:** Sylvia Stamova, Neli Ermenlieva, Gabriela Tsankova, Emilia Georgieva

**Affiliations:** 1Department of Pharmaceutical Chemistry, Faculty of Pharmacy, Medical University of Varna, 9000 Varna, Bulgaria; 2Department of Microbiology and Virusology, Faculty of Medicine, Medical University of Varna, 9000 Varna, Bulgaria; neli.ermenlieva@mu-varna.bg (N.E.); gabriela.tsankova@mu-varna.bg (G.T.); 3Training Sector “Medical Laboratory Technician”, Medical College—Varna, Medical University of Varna, 9000 Varna, Bulgaria; emiliya.georgieva@mu-varna.bg

**Keywords:** lavender essential oil, antimicrobial, adjuvant therapy, in silico study

## Abstract

The increasing prevalence of antimicrobial resistance (AMR) demands novel strategies, including the use of plant-derived agents. This study investigates the chemical profile and in vitro antimicrobial activity of essential oil from *Lavandula angustifolia* (LEO), cultivated in Northeastern Bulgaria. Gas chromatography–mass spectrometry (GC-MS) analysis confirmed the presence of a linalool/linalyl acetate chemotype, characteristic of high-quality lavender oil. LEO demonstrated significant inhibitory activity against *Escherichia coli* ATCC 25922, with minimum inhibitory concentration (MIC) and minimum bactericidal concentration (MBC) values of 0.31% (*v*/*v*) and moderate to weak activity against other Gram-positive and fungal strains. Time–kill assays revealed a concentration-dependent bactericidal effect on *E. coli*. The addition of LEO at subinhibitory concentrations increased the inhibition zones for all antibiotics. In silico analysis identified functional protein clusters potentially modulated by LEO constituents, including targets related to membrane integrity and metabolic regulation. The findings indicate the potential of lavender essential oil as a natural antimicrobial adjuvant; however, additional in vivo and clinical investigations are necessary to validate its therapeutic use. Furthermore, molecular docking analysis revealed a high binding affinity of linalool and linalyl acetate towards the FabI protein of *E.coli*, suggesting a potential inhibitory mechanism at the molecular level.

## 1. Introduction

Antimicrobial resistance (AMR) is increasingly recognized as one of the most critical global health challenges of the 21st century. The emergence and dissemination of multidrug-resistant pathogens—such as *methicillin-resistant Staphylococcus aureus* (MRSA) and extended-spectrum β-lactamase (ESBL)-producing *Escherichia coli*—have significantly undermined the effectiveness of existing antimicrobial therapies, leading to increased morbidity and mortality and substantial economic burdens on healthcare systems [1,2]. The pace of new antibiotic development has failed to keep up with the rapid evolution of resistance, necessitating the exploration of alternative therapeutic approaches [3]. Among these alternatives, essential oils (EOs) extracted from aromatic plants have attracted growing interest as promising antimicrobial agents, owing to their broad-spectrum activity and potential to enhance the efficacy of conventional antibiotics through synergistic interactions [3]. In this context, *Lavandula angustifolia* Mill., the primary botanical source of lavender oil, has demonstrated notable activity against a wide range of pathogenic microorganisms, including resistant strains [4,5,6]. *Lavandula angustifolia* species and hybrids are widely recognized as medicinal plants of considerable economic and therapeutic importance, widely used in the pharmaceutical, cosmetic, food, and aromatherapy industries [7,8,9]. The commercial value and therapeutic efficacy of lavender oil are closely linked to its complex phytochemical composition, predominantly characterized by linalool (20–45%) and linalyl acetate (25–47%), as defined by the European Pharmacopoeia [10] and confirmed by Wińska et al. (2019) [11]. These major constituents are associated with a range of biological effects, including sedative, spasmolytic, antioxidant, and antimicrobial properties [12]. Increasing evidence suggests that LEO can effectively inhibit the growth of resistant bacterial and fungal strains [5].

Beyond its pharmacological potential, lavender cultivation holds significant economic relevance for Bulgaria, which, as of 2022, was the world’s leading producer of lavender with a harvest exceeding 50,000 metric tons [13]. Key cultivars such as “Hemus”, “Druzhba”, and “Karlovo”, developed through extensive breeding programs, exhibit high essential oil yield and strong adaptability to local agroclimatic conditions [14]. However, the expansion of cultivation into novel regions and the growing reliance on generative propagation have led to increased genetic variability, often resulting in chemical inconsistencies in the oil composition [15]. These challenges underscore the pressing need for a comprehensive biochemical and pharmacological characterization of region-specific lavender oils to ensure their therapeutic reliability and industrial standardization.

This study investigates the chemical composition and antimicrobial properties of essential oil derived from *Lavandula angustifolia* cultivar “Jubileina”, grown in the Varna region of Northeastern Bulgaria. Although this region offers favorable agroecological conditions, its impact on the oil’s phytochemical profile and biological activity remains largely unexplored. Most prior research has focused on established lavender-producing areas in South-Central Bulgaria, while northeastern regions like Varna have received limited scientific attention [14,15]. To the best of our knowledge, this is the first study to integrate in vitro evaluation, combination-based antimicrobial testing, and in silico modeling to characterize this chemotype. Building on consistent in vitro evidence of antimicrobial efficacy, we further explored whether key constituents of LEO, when paired with commonly prescribed oral antibiotics, may interact with human pharmacological targets. Such interactions may provide mechanistic insights into the host-mediated modulation of the antimicrobial response. The findings aim to deepen our understanding of the biopharmaceutical relevance of regionally produced lavender oil as a natural source of bioactive compounds, thereby establishing a scientific rationale for future cultivar selection and targeted applications in this part of Bulgaria.

## 2. Results

### 2.1. Essential Oil Extraction

The hydrodistillation of 1200 g of freshly harvested flowering tops of *Lavandula angustifolia* Mill., cultivar “Jubileina,” using a standard Clevenger-type apparatus for 3 h yielded 3.08 mL of essential oil. This corresponds to a yield of 0.26% (*v*/*w*) based on the fresh weight of the plant material. The observed yield falls within the expected range for the “Jubileina” cultivar, which is known to produce lower quantities of essential oil compared to commercial hybrids while exhibiting a superior chemical profile rich in oxygenated monoterpenes such as linalool and linalyl acetate. Reported essential oil yields from fresh *L. angustifolia* typically range between 0.2% and 0.5%, depending on genotype, environmental conditions, and harvest timing [16].

### 2.2. Gas Chromatography–Mass Spectrometry Analysis

The essential oil extracted from the *Lavandula angustifolia* cultivar “Jubileina”, grown in the Varna region of Northeastern Bulgaria, was analyzed by gas chromatography coupled with mass spectrometry (GC-MS). The chromatographic profile revealed a pronounced chemotypic composition dominated by linalool and linalyl acetate, characteristic of the “Linalool/Linalyl Acetate” chemotype, which is widely regarded as an indicator of high-quality lavender oil (Table 1) [5,16]. This chemotype, defined by elevated levels of linalool and linalyl acetate and only trace amounts of camphor and 1,8-cineole, is particularly valued for its pharmacological, aromatherapeutic, and cosmetic properties. The cultivation of *L. angustifolia* in the Varna region benefits from a temperate climate, favorable soil structure, and agroecological conditions that promote the biosynthesis of key terpenoids and esters. Bulgaria continues to be internationally recognized as a leading producer of lavender essential oil, with the region’s environmental factors contributing significantly to the production of aromatic compounds with consistent and desirable phytochemical profiles.

The retention indices (RI) were determined experimentally using a homologous series of n-alkanes (C8–C20) and were compared with literature data obtained under similar chromatographic conditions [17,18]. Minor variations in RI values may result from differences in column polarity or temperature programming. In cases where compounds displayed closely similar RI values (e.g., β-phellandrene and 1,8-cineole), their identification was further confirmed based on distinctive mass spectral patterns and the expected elution order on the polar Stabilwax column.

### 2.3. Determination of Minimum Inhibitory Concentration (MIC) and Minimum Bactericidal/Fungicidal Concentration (MBC/MFC)

The essential oil derived from *L. angustifolia* Mill. cultivar “Jubileina” demonstrated antimicrobial activity against all tested bacterial strains. The minimum inhibitory concentration (MIC) values ranged from 0.31% to 10%, depending on the microorganism. The highest susceptibility was observed in *E. coli* ATCC 25922, for which both the MIC and the minimum bactericidal concentration (MBC) were recorded at 0.31%. In the case of *S. pyogenes* ATCC 12384, the MIC and MBC were determined to be 2.5% and 5%, respectively. Intermediate sensitivity was noted for *S. aureus* ATCC 29213, *K. pneumoniae* ATCC 13883, and *P. aeruginosa* ATCC 27853, with the MIC values ranging between 5% and 10%. *B. subtilis* ATCC 6633 proved to be the most resistant strain, exhibiting both MIC and MBC values at 10%. For the fungal pathogen *C. albicans* ATCC 10231, growth inhibition was observed only at concentrations exceeding 10%; however, no definitive fungicidal effect was detected at the highest concentration tested. All assays included appropriate negative controls (DMSO) and positive controls (standard antibiotics). A summary of these findings is presented in Table 2 and illustrated in Figure 1.

These results confirm the dose-dependent antimicrobial potential of LEO, with the highest sensitivity observed in *E. coli* and limited activity against *C. albicans*.

### 2.4. Time–Kill Kinetics of LEO Against E. coli and C. albicans

In the time–kill assays conducted against *E. coli* ATCC 25922, essential oil from *L. angustifolia* exhibited a clear concentration- and time-dependent bactericidal effect. At the lowest concentration tested (0.31% *v*/*v*), a modest reduction of approximately 1.5 log_10_ CFU/mL in viable cell count was observed over a 24 h period. Increasing the LEO concentration to 0.62% *v*/*v* resulted in a reduction of ≥3 log_10_ CFU/mL by the 8th h, thereby surpassing the threshold for bactericidal activity. At the highest tested concentration (1.24% *v*/*v*), bacterial counts fell below the detection limit (<1 log_10_ CFU/mL) within 6 h, indicating the near-complete eradication of the bacterial population. In contrast, the control sample containing only the solvent exhibited no appreciable decline in viable cell count over the same time period (Figure 2).

In the case of *C. albicans* ATCC 10231, LEO showed markedly lower antifungal efficacy. Even at the highest concentration evaluated (10% *v*/*v*), the reduction in fungal colony-forming units did not exceed 2 log_10_ CFU/mL at 24 h. At concentrations of 5% and 7.5% *v*/*v*, only a minimal inhibition of growth was observed, falling short of the ≥3 log_10_ threshold generally associated with fungicidal activity. Control samples maintained a stable population throughout the experiment, confirming that the limited reductions observed were attributable to the biological activity of the essential oil, rather than any solvent-related effects (Figure 3).

These observations indicate that under the tested conditions, LEO exerts predominantly fungistatic rather than fungicidal activity against *C. albicans.*

### 2.5. Modulatory Effect of Lavender Essential Oil on Antibiotic Activity Against E. coli

Using the disk diffusion method, the addition of LEO at a subinhibitory concentration (0.31% *v*/*v*) consistently enhanced the diameters of inhibition zones for all tested antibiotics against *E. coli* ATCC 25922 (Table 3). The most pronounced increases were observed with ampicillin and gentamicin, while other antibiotics also demonstrated moderate but reproducible enhancements in activity. Notably, LEO alone at 0.31% *v*/*v* produced a consistent inhibition zone measuring approximately 19 mm (mean ± SD, *n* = 3), indicating a stable and reproducible baseline level of antibacterial activity. All differences between antibiotic monotherapy and LEO–antibiotic combinations were statistically significant (*p* < 0.05) across the panel of tested agents. These findings suggest that LEO has the capacity to potentiate antibiotic efficacy, likely through membrane-disruptive properties, increased bacterial cell permeability, or other synergistic mechanisms. Although preliminary, these results highlight the potential of natural products such as LEO to serve as adjunctive agents in antimicrobial therapy, supporting further investigation into their role in enhancing conventional treatments.

The most pronounced increase in the inhibition zone was observed with ampicillin and gentamicin, while the remaining agents exhibited a moderate yet consistent enhancement in antimicrobial activity. Notably, LEO alone at the tested subinhibitory concentration demonstrated stable antimicrobial efficacy. Statistical analysis confirmed that the differences between monotherapy and combination treatments were significant for all tested agents. These findings suggest a potential modulatory effect of the essential oil, manifesting as an amplification of antibiotic efficacy—likely attributable to membrane-disruptive actions, increased cellular permeability, or other complementary mechanisms of action. Although preliminary, these observations underscore the potential utility of natural products such as LEO as adjunctive agents in antimicrobial therapy.

### 2.6. In Silico Target Prediction and Protein–Protein Interaction (PPI) Network Analysis

Given the potential of LEO to enhance antibacterial efficacy through the modulation of both microbial and host-mediated mechanisms, we conducted an in silico analysis of predicted human protein targets for its major constituents—linalool and linalyl acetate—alongside selected antibiotics (ampicillin, ceftriaxone, meropenem, gentamicin, and ciprofloxacin). Predicted targets were analyzed using the STRING v11.5 database to construct protein–protein interaction (PPI) networks, and functional modules were subsequently identified using Markov Cluster Algorithm (MCL) clustering.

#### 2.6.1. LEO and Ampicillin

We constructed a protein–protein interaction (PPI) network using STRING v11.5 to explore the molecular mechanisms underlying the biological activity of linalool, linalyl acetate, and ampicillin. This analysis provided an overview of predicted interactions among the identified human protein targets and revealed several functionally distinct clusters. These included proteins associated with proteolysis and extracellular matrix remodeling, ion transport and pH regulation, cell signaling and proliferation, and neuromodulation. Central nodes such as EGFR and DRD2 were identified as potential key regulators within the network. These findings suggest that components of lavender essential oil (LEO) may influence not only antimicrobial activity but also broader physilogical functions, particularly those related to neural regulation (Figure 4).

The resulting PPI network revealed four major functional clusters: (1) proteolysis and extracellular matrix remodeling, including cathepsins and elastase; (2) ion transport and pH regulation; (3) cell signaling and proliferation, with epidermal growth factor receptor (EGFR) emerging as a central hub; and (4) neuromodulation, comprising neurotransmitter receptors. These findings suggest that LEO may exert systemic neurophysiological effects in addition to its antimicrobial properties.

#### 2.6.2. LEO and Ceftriaxone

A similar modular organization was observed, with clusters involving enzymes linked to inflammatory responses, metabolic regulation, and intracellular signaling. EGFR again occupied a key network position. The presence of neuromodulatory receptors within the network suggests that systemic physiological modulation by LEO may contribute to the observed synergistic effects when combined with ceftriaxone.

#### 2.6.3. LEO and Meropenem

The interaction network displayed several distinct clusters: (1) GABAergic, dopaminergic, and opioid receptors; (2) adrenergic receptors involved in metabolic and cardiovascular regulation; (3) inflammatory signaling proteins and steroid-responsive factors, including central nodes such as EGFR and NR3C1; (4) proteolytic enzymes; (5) ion channels and sensory receptors; and (6) immunomodulatory molecules. The structured nature of this network indicates potential for synergistic action through the engagement of diverse host physiological systems.

#### 2.6.4. LEO and Gentamicin

To investigate the potential molecular pathways influenced by the combination of linalool, linalyl acetate, and gentamicin, we generated a protein–protein interaction (PPI) network using the STRING v11.5 platform. This network analysis offered a comprehensive overview of predicted interactions among human target proteins and revealed several biologically relevant functional clusters. These included proteins related to neuromodulation, adrenergic signaling, inflammatory processes, and proteolysis. Central hubs such as EGFR, NR3C1, and PARP1 were identified as key regulatory nodes. Additionally, targets associated with acid–base homeostasis, lipid metabolism, and immune responses were detected. The results support the hypothesis that lavender essential oil (LEO), in conjunction with gentamicin, may exert wide-ranging host-modulatory effects beyond its antimicrobial properties (Figure 5).

The PPI network for this combination revealed functional clusters consistent with previous pairings, including neuromodulation, adrenergic signaling, inflammatory pathways, and proteolysis. EGFR, NR3C1, and PARP1 emerged as key hubs. Additional components involved in acid–base regulation, lipid metabolism, and immune function were identified, suggesting a broad host-modulating capacity for LEO in conjunction with gentamicin.

#### 2.6.5. LEO and Ciprofloxacin

The network analysis identified a complex structure with central nodes such as NR3C1, PI3K isoforms, and GSK3B, which are involved in inflammation, cell survival, and immune signaling. Distinct clusters were also observed for neurotransmitter receptors, cathepsins, carbonic anhydrases, integrins, and proteins involved in steroid metabolism. The overlap between LEO and ciprofloxacin targets suggests a complementary regulatory influence on host responses, potentially underpinning enhanced therapeutic efficacy.

Across all combinations, the PPI networks demonstrated considerable functional complementarity between the predicted human targets of LEO and those of conventional antibiotics. The most consistently represented clusters were associated with inflammatory regulation, cellular proliferation, ion transport, and neuromodulation. Central nodes such as EGFR and NR3C1 recurred in multiple networks, underscoring their potential as convergence points for LEO-mediated modulation. These findings support the hypothesis that LEO may potentiate antibiotic action not only through direct antimicrobial activity but also via the favorable modulation of host physiological pathways.

These converging pathways suggest that the therapeutic benefit of LEO may extend beyond direct antimicrobial action to the modulation of host immune, metabolic, and signaling responses.

### 2.7. Docking of Linalool and Linalyl Acetate to FabI

Molecular docking analysis revealed that both linalool and linalyl acetate favorably bind within the NADH binding cleft of *E. coli* FabI. Notably, linalyl acetate demonstrated consistently stronger affinity, with the top-ranked binding pose exhibiting a ΔG of –5.98 kcal/mol and FullFitness of –17.04, compared to linalool (ΔG: –5.19 kcal/mol; FullFitness: –8.49).

As shown in Table 4, linalyl acetate maintained superior binding energies across all top five docked conformations. Visual analysis (Figure 6 indicated that linalyl acetate engages in key hydrogen bonding and hydrophobic interactions with residues Tyr156 and Lys163, suggesting the stabilizing role of its ester moiety. Conversely, linalool showed a more limited interaction profile, primarily via van der Waals contacts.

## 3. Discussion

The present study provides an interdisciplinary analysis of the antimicrobial potential of LEO, evaluated both as a single agent and in combination with conventional antibiotics. This work integrates validated in vitro antimicrobial assays with complementary in silico predictions, aiming to elucidate not only the direct effects of LEO on microbial viability but also its potential interactions at the host level. Together, these findings contribute to the broader understanding of plant-derived compounds as prospective therapeutic adjuvants in the context of antimicrobial resistance.

LEO demonstrated significant antimicrobial activity against a broad spectrum of pathogenic microorganisms, including Gram-negative bacteria such as *E. coli*. In contrast, its activity against the fungal pathogen *C. albicans* was limited, not exceeding a fungistatic effect under the tested conditions. The lack of fungicidal activity against *C. albicans* is consistent with previous reports and may reflect inherent differences in membrane composition or efflux mechanisms. Although the outer membrane of Gram-negative bacteria typically confers resistance to hydrophobic agents such as essential oils, previous studies have demonstrated the susceptibility of *E. coli* to LEO [19,20,21]. The MIC and MBC values reported in the literature vary between 0.25% and 1% (*v*/*v*), depending on the strain and experimental conditions [21,22]. For example, Hossain et al. (2017) observed MIC and MBC values ranging from 0.5% to 2% for Gram-negative isolates from turtles, indicating a strong bactericidal effect (MBC/MIC < 4) [22]. Similarly, Ratajczak et al. (2023) reported an MIC of 500 µg/mL against *E. coli* ATCC 25922 [23].

Increasing attention has been directed towards the capacity of essential oils to enhance the efficacy of antibiotics, particularly against resistant Gram-negative strains. Yang et al. (2020) demonstrated that combining LEO with meropenem resulted in a 15-fold reduction in the effective dose of the oil and a 4-fold reduction in the MIC of meropenem against carbapenemase-producing *K. pneumoniae* [24]. Comparable results have been reported for combinations of LEO with ampicillin and gentamicin, where significantly larger inhibition zones were observed relative to the antibiotics alone [25,26,27]. These in vitro findings corroborate the observed enhancement in antibacterial efficacy when LEO is administered alongside conventional antibiotics, particularly ampicillin and gentamicin. The consistent increase in inhibition zone diameters supports the hypothesis that LEO may facilitate enhanced antibiotic penetration or exert additional, complementary mechanisms of action.

Donadu et al. (2018) similarly reported the improved efficacy of gentamicin when co-administered with LEO against Gram-negative pathogens [25]. These synergistic effects were extensively reviewed by Langeveld et al. (2014), who highlighted multiple mechanisms by which essential oils may potentiate antibiotic action [28].

Mechanistically, the antimicrobial effect of LEO is primarily attributed to its disruptive impact on bacterial membranes. Its major constituents—linalool and linalyl acetate—are known to compromise membrane integrity and fluidity, leading to the leakage of intracellular contents, the dissipation of the proton motive force, and, ultimately, bacterial cell death [24,29]. Electron microscopy studies have confirmed morphological damage to bacterial envelopes and the increased uptake of hydrophobic compounds following treatment with LEO [24,25]. Notably, antimicrobial activity in the vapor phase has also been documented, including against methicillin-resistant *S. aureus* (MRSA) and various Gram-negative strains [24,30].

Beyond physical membrane disruption, LEO has been shown to elevate the intracellular levels of reactive oxygen species (ROS), contributing to oxidative stress within bacterial cells and enhancing their susceptibility to antibiotics [20,27]. Exposure to LEO results in increased lipid peroxidation and decreased membrane protein expression, further reinforcing its bactericidal effects [24,31]. In silico molecular profiling has identified bacterial targets such as FabH and SasG, which are involved in lipid biosynthesis and adhesion, respectively—functions that, when inhibited, may reduce both bacterial virulence and resistance [26,32].

Moreover, an analysis of predicted human protein targets revealed that both LEO constituents and selected antibiotics may influence overlapping regulatory and signaling pathways. Protein–protein interaction networks generated using the STRING platform revealed clusters associated with immune regulation (e.g., EGFR, IDO1, PARP1), extracellular matrix remodeling (e.g., cathepsins), and neuromodulation (e.g., GABA and adrenergic receptors) [33]. These host-level interactions suggest that part of LEO’s antimicrobial benefit may derive from indirect effects, such as the modulation of inflammation, barrier integrity, or cellular stress responses, in addition to its direct antimicrobial actions. These proposed host-med iated mechanisms are consistent with the known biological activities of LEO, which include anti-inflammatory, antioxidant, and immunomodulatory effects. Both in vitro and in vivo studies have demonstrated that LEO can downregulate pro-inflammatory cytokines such as IL-1β and TNF-α while enhancing macrophage phagocytic capacity in models of *S. aureus* infection [34]. Systemic administration in rodent models has been associated with decreased oxidative stress and improved tissue histology [35]. Furthermore, LEO has been employed in antimicrobial food preservation and packaging materials, leading to significant reductions in microbial contamination [36].

Altogether, the present findings support the potential of *L. angustifolia* essential oil as a multifunctional antimicrobial agent. Its activity encompasses both direct bactericidal effects and the modulation of host pathways, indicating potential utility in combination therapies and alternative delivery formats. The antimicrobial efficacy of the essential oil derived from *L. angustifolia* cultivated in Northeastern Bulgaria (cultivar “Jubileina”) aligns closely with previously published data on lavender oils from other Bulgarian regions. Mladenova et al. (2024) reported MIC values below 0.1125% (*v*/*v*) against *E. coli*, *S. aureus*, *S. enterica*, and *C. albicans*, which are comparable to the MIC of 0.31% observed against *E. coli* in the present study [37].

While geographic and environmental variables—such as soil composition, altitude, and local microclimate—are known to influence essential oil composition, Bulgarian lavender oils consistently exhibit high concentrations of linalool and linalyl acetate, the primary constituents responsible for their antimicrobial activity [11,26]. According to the current edition of the *European Pharmacopoeia* (11.0, 2023), authentic *L. angustifolia* oil must contain 20–45% linalool and 25–47% linalyl acetate [10]. The oil analyzed in the present study conforms to these specifications in nearly all respects, with 29.94% linalool and 24.31% linalyl acetate. The slight deviation below the minimum linalyl acetate threshold may reflect microclimatic conditions specific to the region or variability in harvest timing. Importantly, the levels of camphor (0.43%) and 1,8-cineole (1.10%) remain well within pharmacopeial limits, confirming the high chemical quality of the product.

Comparative studies from Southern Bulgaria (e.g., Karlovo, Kazanlak) have reported similar antimicrobial efficacy against both Gram-positive and Gram-negative strains, with MIC values typically ranging from 0.25% to 1.0% (*v*/*v*) [23,26,38]. These data suggest that, provided phytochemical quality standards are met, geographic origin alone does not significantly influence the antimicrobial potential of Bulgarian lavender oil. The present findings thus support the broader conclusion that high-grade *L. angustifolia* oils from diverse Bulgarian regions—including the relatively underexplored northeast—exhibit consistent and therapeutically relevant bioactivity.

Beyond the observed in vitro antimicrobial effects, our in silico analysis provides further insight into the mechanisms by which the main constituents of lavender essential oil exert their activity. Linalool and linalyl acetate have been shown to interact with bacterial membrane-associated proteins, metabolic enzymes, and quorum-sensing regulators. Prior studies have demonstrated that linalool disrupts the cytoplasmic membrane of *Pseudomonas fluorescens*, resulting in the leakage of intracellular contents and the inhibition of key metabolic enzymes such as succinate dehydrogenase and ATP synthase, ultimately leading to ATP depletion and oxidative imbalance [39]. Similar membrane damage and oxidative stress have been reported in *Shigella sonnei* [40], while metabolomic analyses in *Pseudomonas fragi* have confirmed decreased amino acid levels and disrupted redox homeostasis following linalool exposure [41].

In parallel with these direct effects on bacteria, our molecular modeling results suggest interactions between essential oil constituents and human proteins involved in inflammation, oxidative stress, and epithelial barrier function. Among the most relevant targets identified in our protein–protein interaction network were EGFR, PARP1, and NR3C1. EGFR plays a key role in epithelial repair and immune modulation; NR3C1, the glucocorticoid receptor, regulates inflammatory responses through the suppression of cytokine expression; and PARP1 is involved in DNA repair and the regulation of inflammatory gene expression. Previous studies have shown that lavender essential oil can modulate these pathways, thereby reducing inflammation and oxidative damage [42,43].

The inclusion of molecular docking analysis in the present study was prompted by the need to elucidate the potential direct interactions between key constituents of *Lavandula angustifolia* essential oil (LEO) and validated microbial targets. Among the tested microorganisms, *Escherichia coli* ATCC 25922 exhibited the highest susceptibility to LEO, with both minimum inhibitory concentration (MIC) and minimum bactericidal concentration (MBC) values recorded at 0.31% (*v*/*v*). This finding, coupled with the clinical relevance of *E. coli* as a Gram-negative pathogen frequently associated with multidrug-resistant infections, particularly extended-spectrum β-lactamase (ESBL)-producing strains, justified its selection as a model organism for mechanistic exploration [44].

To investigate the antibacterial mode of action at the molecular level, we selected the enoyl-acyl carrier protein reductase FabI as the docking target. FabI is a central enzyme in bacterial fatty acid biosynthesis and has been extensively studied as a druggable target in *E. coli*, due to its role in maintaining membrane integrity and viability [45,46]. Its high-resolution crystal structure (PDB ID: 4CV3) enables the accurate modeling of ligand–target interactions and supports computational approaches to structure-based antimicrobial discovery. Docking simulations revealed that both linalool and linalyl acetate, the two major monoterpenes in LEO, bind within the NADH-binding pocket of FabI. Linalyl acetate displayed stronger predicted binding affinity, forming additional polar interactions with key active site residues such as Tyr156 and Lys163. In contrast, linalool, which lacks an ester moiety, engaged primarily in hydrophobic contacts and exhibited weaker overall interaction energy. These results are in line with prior studies demonstrating that the presence of polar functional groups in monoterpenes enhances their affinity for FabI and related enzymes through hydrogen bonding and electrostatic stabilization [46]. The observed binding patterns underscore the structural relevance of esterification in modulating ligand–target interactions and suggest that linalyl acetate may act as a more effective inhibitor of FabI. This may contribute to the observed bactericidal effects of LEO against *E. coli* and supports the hypothesis that direct enzymatic inhibition is at least partially responsible for its antimicrobial activity.

Taken together with the time–kill assay data and the demonstrated synergistic interactions between LEO and selected antibiotics, the docking findings reinforce the proposed multifactorial mechanism of action of LEO. While further in vivo studies are necessary to validate the pharmacological significance of FabI inhibition, the present results provide a strong rationale for including docking-based molecular evaluations in the study of natural antibacterial agents.

## 4. Materials and Methods

### 4.1. Reagents, Media, and Antibiotics

All reagents used in this study were of analytical grade. Dimethyl sulfoxide (DMSO, ≥99.5%, Sigma-Aldrich, Darmstadt, Germany) was used as the solvent for preparing essential oil stock solutions. Anhydrous sodium sulfate (Na_2_SO_4_, Ph Eur, Merck, Darmstadt, Germany) was used to dry the oil following distillation. Mueller–Hinton broth and Brain Heart Infusion broth served as culture media for bacterial and fungal strains, respectively. For plating and CFU determination, blood agar (used for MBC/MFC testing), nutrient agar (for *E.coli*), and Sabouraud dextrose agar (for *C. albicans*) were employed. Serial dilutions were prepared using sterile physiological saline. Antibiotic susceptibility testing was performed using commercially available disks containing ampicillin (10 µg), ceftriaxone (30 µg), meropenem (10 µg), ciprofloxacin (5 µg), and gentamicin (10 µg), in accordance with Clinical and Laboratory Standards Institute (CLSI) guidelines [47].

### 4.2. Essential Oil Extraction

Plant material from *Lavandula angustifolia* Mill., cultivar “Jubileina,” was grown in the Varna region of Northeastern Bulgaria, near the village of Barite (43°14′55″ N, 27°38′12″ E). At the time of harvest in July 2023, the plants were three years old and cultivated without the use of mineral fertilizers, supplemental irrigation, or chemical pesticides, relying entirely on natural rainfall. Harvesting was performed manually in the early morning under dry weather conditions during peak flowering, with only the inflorescences collected to ensure a high content of volatile compounds.

The investigated material represents a commercial population of the “Jubileina” cultivar, selected through breeding rather than wild collection. As its identity is well established within regional agricultural practice and confirmed on site, a herbarium specimen was not deposited.

Essential oil was extracted from fresh plant material by steam distillation. Following condensation, the aqueous layer was removed, and the oil was collected by decantation, dried over anhydrous sodium sulfate, and stored in amber glass vials at 4 °C. The obtained volume was sufficient for all analyses, including MIC and MBC/MFC testing, time–kill assays, and GC–MS profiling, each performed in triplicate. Aliquots were prepared under sterile conditions and stored under the same conditions to prevent oxidation and maintain the integrity of the volatile constituents.

### 4.3. Chemical Composition Analysis

The chemical composition of *L. angustifolia* essential oil was analyzed using a gas chromatograph (Agilent Technologies 7890A, Santa Clara, CA, USA) fitted with a flame ionization detector (FID) and a mass spectrometric detector (5975C MSD, Santa Clara, CA, USA). A Stabilwax capillary column (Restek, Bellefonte, PA, USA; 30 m length × 0.25 mm i.d., 0.25 µm film thickness) was used for the analysis. The oven temperature was initially set at 65 °C and programmed to increase to 170 °C at a rate of 1.5 °C/min, with a total run time of 70 min. Injector and detector temperatures were maintained at 250 °C. Hydrogen and helium were employed as carrier gases at a constant flow rate of 0.8 mL/min. The mass spectrometer operated in scan mode over an *m*/*z* range of 40–450. Samples (1.0 µL) were injected in split mode using a 100:1 split ratio. The identification of volatile components was based on a combined approach, including Kovats retention indices (RI) and full mass spectral matching using the NIST’08 [17] and Adams [18] libraries. When two compounds showed similar or overlapping RI values, their identification was carefully confirmed based on their distinct mass spectral fragmentation patterns and the expected elution order on the chromatographic column, reducing the risk of misidentification due to co-elution. Such overlaps are well-documented in the GC-MS analysis of essential oils and typically reflect the co-elution of structurally related compounds [48,49].

### 4.4. Determination of Minimum Inhibitory Concentration (MIC) and Minimum Bactericidal/Fungicidal Concentration (MBC/MFC)

The antimicrobial activity of *L. angustifolia* essential oil was evaluated using the broth microdilution method, following established standard procedures. This study included reference microbial strains obtained from the American Type Culture Collection (ATCC, provided by Ridacom, Bulgaria): *Escherichia coli* ATCC 25922, *Staphylococcus aureus* ATCC 29213, *Bacillus subtilis* ATCC 6633, *Klebsiella pneumoniae* ATCC 13883, Pseudomonas aeruginosa ATCC 27853, *Streptococcus pyogenes* ATCC 12384, and the fungal strain *Candida albicans* ATCC 10231. A 10% (*v*/*v*) stock solution of the essential oil was prepared in 1% Dimethyl sulfoxide (DMSO). Serial two-fold dilutions were performed using Mueller–Hinton broth (MHB) for bacterial strains and Brain Heart Infusion (BHI) broth for the fungal strain, yielding final concentrations ranging from 10% to 0.078% (*v*/*v*). Each well was inoculated with 0.1 mL of microbial suspension, adjusted to a turbidity equivalent to the 0.5 McFarland standard (~1 × 10^6^ CFU/mL), obtained from 4–6 h broth cultures. Bacterial strains were incubated at 37 °C for 24 h, while *C. albicans* was incubated at 35 °C for 48 h. The minimum inhibitory concentration (MIC) was defined as the lowest concentration of essential oil that resulted in no visible growth. To determine the minimum bactericidal concentration (MBC) or minimum fungicidal concentration (MFC), aliquots from wells without visible turbidity were plated on blood agar and incubated under the same conditions. The MBC and MFC were defined as the lowest concentrations at which no colony formation was observed. All experiments were performed in triplicate.

### 4.5. Time–Kill Kinetics of LEO Against E. coli and C. albicans

The dynamic antimicrobial efficacy of *L. angustifolia* essential oil (LEO) was evaluated through time–kill assays against the reference strains *Escherichia coli* ATCC 25922 and *Candida albicans* ATCC 10231. The methodology was based on the guidelines of CLSI document M26-A for bacteria and M27-A for yeasts, with appropriate modifications to account for the physicochemical properties of the test substance. Inocula were prepared from 4–6 h broth cultures and adjusted with sterile saline to an optical density equivalent to the 0.5 McFarland standard, corresponding to approximately 1 × 10^6^ CFU/mL. LEO was emulsified in 1% Dimethyl sulfoxide (DMSO) to achieve the desired test concentrations. For E. coli, concentrations of 0.31%, 0.62%, and 1.24% (*v*/*v*) were tested; for *C. albicans*, concentrations of 5%, 7.5%, and 10% (*v*/*v*) were applied. Control groups consisted of microbial inocula in 1% DMSO without LEO. All experimental mixtures were incubated at 37 °C with continuous shaking at 150 rpm. At predetermined time points (0, 2, 4, 6, 8, and 24 h), 100 µL aliquots were withdrawn, serially diluted in sterile saline, and surface-plated on appropriate agar media—nutrient agar for *E. coli* and Sabouraud dextrose agar for *C. albicans*. Colony-forming units (CFU) were enumerated following 24 h of incubation at 37 °C. The results were expressed in logarithmic form as log_10_ (CFU/mL). A ≥3 log_10_ reduction in viable cell count compared to the initial inoculum was interpreted as indicative of bactericidal or fungicidal activity. All experiments were conducted in three independent biological replicates, with technical duplicates for each condition.

### 4.6. Modulatory Effect of Lavender Essential Oil on Antibiotic Activity Against E. coli

The potential interaction between LEO and selected antimicrobial agents was evaluated using a modified disk diffusion method. The assay was performed against the reference strain E. coli ATCC 25922, cultured on Mueller–Hinton agar in accordance with CLSI guidelines. A microbial inoculum was prepared from a fresh 18 h culture and adjusted to a turbidity equivalent to the 0.5 McFarland standard. A volume of 100 µL of the standardized suspension was evenly spread across the surface of the agar plates, which were then allowed to dry at room temperature for 10–15 min. The antimicrobial agents tested included antibiotic disks containing ampicillin (10 µg), ceftriaxone (30 µg), meropenem (10 µg), ciprofloxacin (5 µg), and gentamicin (10 µg). LEO was applied in two ways: 10 µL of pure essential oil was placed on sterile blank disks to assess the effects of LEO alone, and the same volume was pipetted directly onto the pre-impregnated antibiotic disks to evaluate potential synergistic interactions. Control disks included antibiotics alone and LEO alone. After disk placement, the plates were incubated for 18–24 h at 37 °C under aerobic conditions. Inhibition zone diameters were measured in millimeters using a calibrated ruler. All experiments were conducted in triplicate, and the mean inhibition zone values were calculated for each condition.

### 4.7. In Silico Target Prediction and PPI Network Analysis

Following the confirmation of LEO’s antimicrobial activity and its observed enhancement in conventional antibiotic effects in vitro, we extended our investigation to explore potential interactions between its main constituents (linalool and linalyl acetate) and human protein targets. Additionally, five commonly used orally administered antibiotics—ampicillin, ceftriaxone, meropenem, ciprofloxacin, and gentamicin—were included in the analysis. The aim was to identify potential pharmacological overlaps at the host level that may be relevant to infection dynamics, immune modulation, or tissue response. For each compound, chemical structures were analyzed using the UniProtKB/Swiss-Prot database and the SwissTargetPrediction platform to predict human protein targets [50,51]. The top 15 targets for each compound were selected and further examined using the STRING database to construct protein–protein interaction (PPI) networks, with an emphasis on functionally related clusters involved in metabolic regulation, immune signaling, and cellular stress responses [52].

### 4.8. Molecular Docking Protocol

The three-dimensional structure of *E. coli* enoyl-acyl carrier protein reductase (FabI) was retrieved from the Protein Data Bank (PDB ID: 4CV3). The protein structure was prepared using AutoDock Tools 1.5.6 by removing water molecules and non-relevant heteroatoms while retaining NADH as a cofactor. Polar hydrogens were added, and Gasteiger charges were computed. The ligands, linalool and linalyl acetate, were obtained in 3D format and minimized using MMFF94 force fields. Docking was performed using SwissDock in accurate mode with default settings. The binding site was defined around the NADH binding pocket, based on co-crystallized ligand coordinates. The resulting docked conformations were evaluated based on Full Fitness scores and the estimated binding free energy (ΔG) [53].

### 4.9. Statistical Analysis

All experiments were performed in triplicate (*n* = 3). MIC and MBC/MFC values are reported as the mean ± standard deviation (SD), with no statistical tests applied to these measurements. Differences in inhibition zone diameters between antibiotic treatments with and without LEO were assessed using two-tailed Student’s *t*-tests, considering *p* < 0.05 as the threshold for significance. All analyses were carried out using Jamovi (version 2.4.8) [54]. Because of the small sample size (*n* = 3), data normality was not formally evaluated, and the results were interpreted descriptively rather than confirmatively.

## 5. Conclusions

The essential oil of *Lavandula angustifolia*, extracted from the “Jubileina” cultivar cultivated in the Varna region of Northeastern Bulgaria, exhibited significant antimicrobial activity against reference pathogenic microorganisms, with particularly pronounced efficacy against *E. coli*. The demonstrated synergistic effects with several clinically relevant antibiotics, supported by in silico analyses, suggest the presence of complementary mechanisms of action. The observed modulatory effects are likely attributable to disruptions in membrane integrity, the induction of oxidative stress, and interference with key metabolic and signaling pathways. While these findings are encouraging, they should be considered preliminary. Further in vivo and clinical investigations are warranted to comprehensively assess the efficacy, safety, and therapeutic potential of lavender essential oil as an adjuvant in antimicrobial treatment strategies.

## Figures and Tables

**Figure 1 antibiotics-14-00656-f001:**
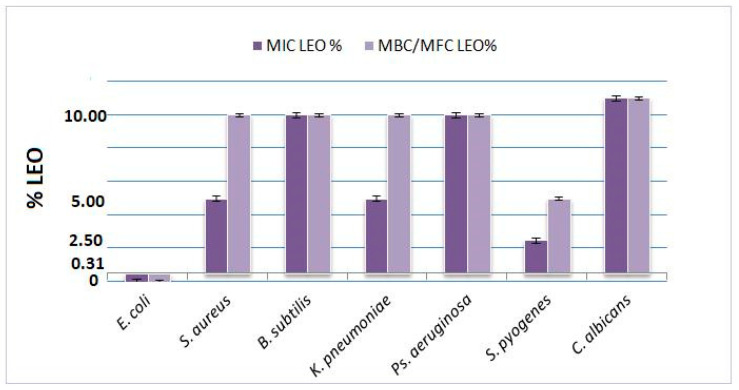
Mean MIC and MBC/MFC values (% *v*/*v*) of LEO against tested microorganisms (*n* = 3). Error bars represent standard deviation.

**Figure 2 antibiotics-14-00656-f002:**
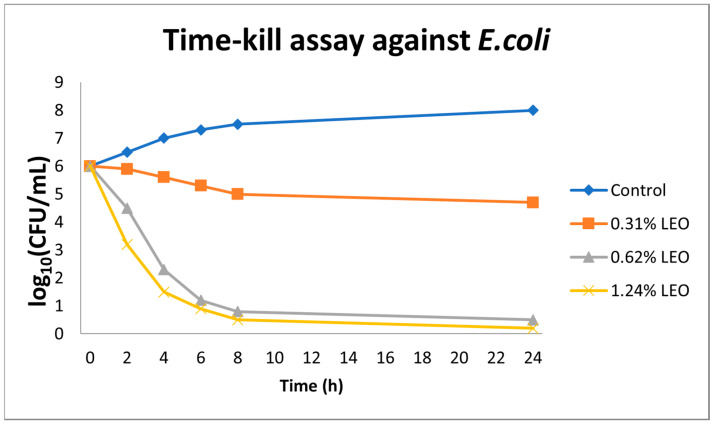
Time–kill kinetics of *Escherichia coli* exposed to lavender essential oil (LEO) at concentrations of 0.31%, 0.62%, and 1.24% (*v*/*v*) over 24 h. Values represent mean log_10_ CFU/mL ± SD (*n* = 3). Data are descriptive; no statistical testing was applied.

**Figure 3 antibiotics-14-00656-f003:**
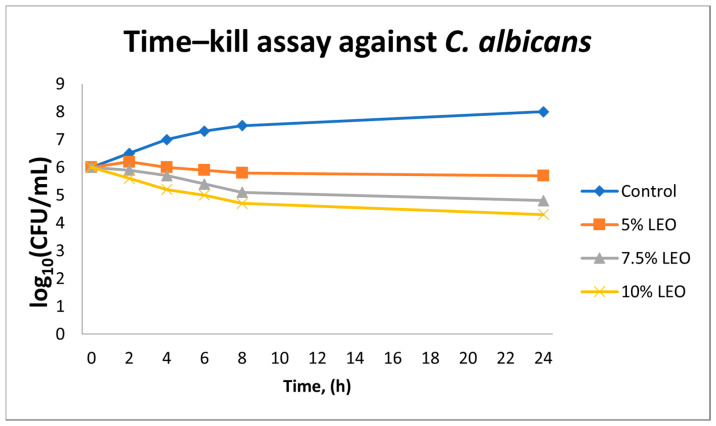
Time–kill kinetics of *Candida albicans* exposed to lavender essential oil (LEO) at concentrations of 5%, 7.5%, and 10% (*v*/*v*) over 24 h. Values represent mean log_10_ CFU/mL ± SD (*n* = 3). Data are descriptive; no statistical testing was applied.

**Figure 4 antibiotics-14-00656-f004:**
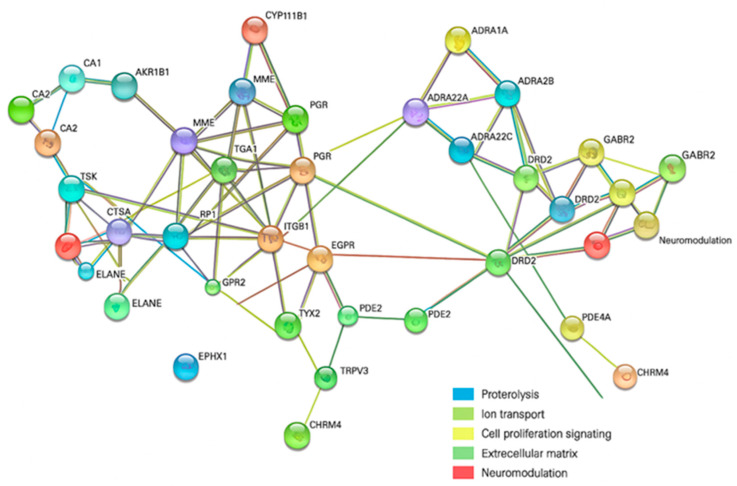
Protein–protein interaction (PPI) network of predicted human targets of linalool, linalyl acetate, and ampicillin, constructed using STRING v11.5. Nodes represent proteins; edges and cluster colors indicate interaction types and functional groupings (see legend in figure).

**Figure 5 antibiotics-14-00656-f005:**
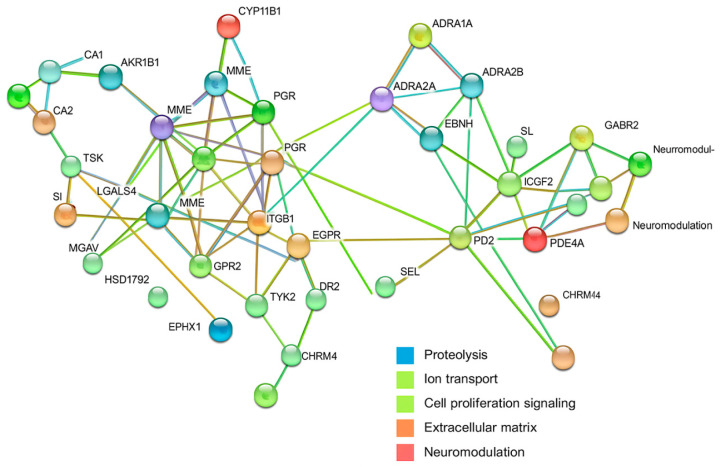
Protein–protein interaction (PPI) network of predicted human targets of linalool, linalyl acetate, and gentamicin, constructed using STRING v11.5. Nodes represent proteins; edges and cluster colors indicate interaction sources and biological functions (see legend in figure).

**Figure 6 antibiotics-14-00656-f006:**
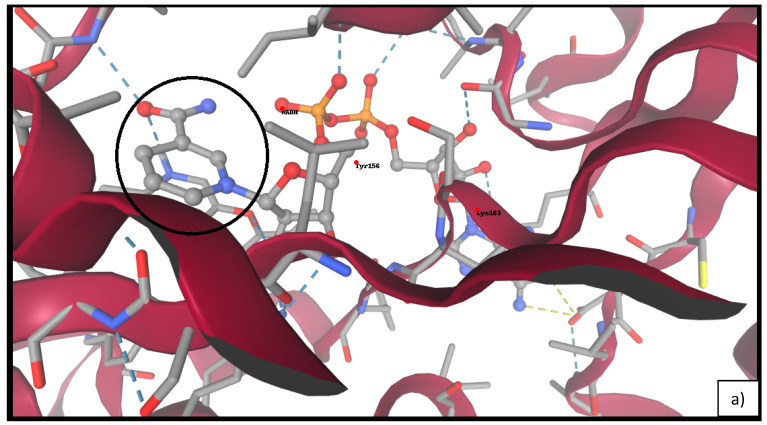
Molecular docking of FabI (PDB ID: 4CV3) with linalool (**a**) and linalyl acetate (**b**). The black circles indicate the binding site regions and key molecular interactions in each protein–ligand complex.

**Table 1 antibiotics-14-00656-t001:** Chemical composition of *Lavandula angustifolia* essential oil.

Lavender EO Compounds	Mean % Area (*n* = 3)	Kovats Index
α-Pinene	0.55	939
Camphene	0.32	954
β-Pinene	0.17	980
1-Octen-3-ol	0.49	974
3-Octanone	1.52	987
β-Myrcene	0.95	988
β-Phellandrene	0.18	1033
cis-β-Ocimene	7.22	1029
1,8-cineol	1.10	1033
Limonene	0.24	1031
trans-β-Ocimene	5.30	1042
Linalool	29.94	1100
1-Octen-3-yl-acetate	0.93	1100
Hexyl butyrate	0.33	1165
Camphor	0.43	1146
Borneol	0.84	1165
Terpinene-4-ol	5.20	1174
α-Terpineol	1.58	1186
Linalyl acetete	24.31	1250
Lavandulol acetate	3.25	1290
Lavandulol	1.07	1418
β-Caryophyllene	3.94	1582
	89.96%	

**Table 2 antibiotics-14-00656-t002:** Minimum inhibitory (MIC) and bactericidal/fungicidal (MBC/MFC) concentrations of lavender essential oil (LEO) against tested microorganisms.

Microorganism	MIC LEO %(*v*/*v*) ± SD	MBC/MFC LEO%(*v*/*v*) ± SD
*E. coli*	0.31 ± 0.00	0.31 ± 0.00
*S. aureus*	5.00 ± 0.00	10.00 ± 0.00
*B. subtilis*	10.00 ± 0.00	10.00 ± 0.00
*K. pneumoniae*	5.00 ± 0.29	10.00 ± 0.00
*P. aeruginosa*	10.00 ± 0.00	10.00 ± 0.00
*S. pyogenes*	2.50 ± 0.29	5.00 ± 0.50
*C. albicans*	>10.00	>10.00

Note: Values represent mean ± SD from three independent experiments (*n* = 3). No statistical tests were applied. For *C. albicans*, no complete inhibition was observed within tested range (>10%).

**Table 3 antibiotics-14-00656-t003:** Modulatory effect of lavender essential oil (0.31% *v*/*v*) on antimicrobial activity of selected antibiotics against *Escherichia coli.*

Antimicrobial Agent	Antibiotic, mm ± SD	LEO Alone 0.31%, mm ± SD	Antibiotic + LEO, mm ± SD	*p*-Value
Ampicillin (A)	23.00 ± 0.50	19.00 ± 0.00	31.00 ± 0.60	0.0002
Ceftriaxone (CRO)	34.00 ± 0.30	19.00 ± 0.00	38.00 ± 0.40	0.0265
Meropenem (MEM)	38.00 ± 0.40	19.00 ± 0.00	41.00 ± 0.50	0.0047
Ciprofloxacin (CIP)	45.00 ± 0.20	19.00 ± 0.00	48.00 ± 0.30	0.0079
Gentamicin (GEN)	25.00 ± 0.40	19.00 ± 0.00	32.00 ± 0.60	0.0008

Note: Values are presented as mean ± SD from three independent replicates. Statistical analysis was performed using two-tailed Student’s *t*-test.

**Table 4 antibiotics-14-00656-t004:** Molecular docking results of linalool and linalyl acetate with FabI–protein (PDB ID: 4CV3).

	Linalool FullFitness	Linalool Score (kcal/mol)	Linalyl Acetate FullFitness	Linalyl Acetate (kcal/mol)
1.	−1746.45	−7.45	−1872.78	−8.93
2.	−1744.13	−7.38	−1871.35	−8.82
3.	−1743.01	−7.35	−1869.45	−8.76
4.	−1741.89	−7.33	−1868.02	−8.71
5.	−1741.20	−7.31	−1866.89	−8.64

Note: All binding poses were generated using SwissDock with full flexibility; docking scores represent predicted binding free energies (ΔG) in kcal/mol.

## Data Availability

All data generated or analyzed during this study are included in the published article.

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
