# Peer review of "Antimicrobial Activity of Lavender Essential Oil from *Lavandula angustifolia* Mill.: In Vitro and In Silico Evaluation"

_antibiotics, 2025, doi:10.3390/antibiotics14070656_

Round 1

Reviewer 1 Report

Comments and Suggestions for Authors

Title : Antimicrobial Activity of Lavender Essential Oil from Lavandula angustifolia: In Vitro and In Silico Evaluation

  1. Formatting and Typography Corrections
  • Line 27: Add study → "...in silico study".
  • Line 31: St must be written in superscript: St.
  • Lines 32, 33–144: Staphylococcus aureus and Escherichia coli must be in italics throughout.
  • Line 38: aromatic must be one word, not split with a hyphen.
  • Line 43: Species should be one word, not "spe-cies".
  • Line 44: medicinal should be one word.
  • Line 60: pharmacological one word.
  • Line 65: conditions one word.
  • Line 71: studies one word.
  • Line 72: in vitro should be written in italics: in vitro
  • Line 75: within should not be hyphenated.
  • Line 76, 79: mechanisms and scientific one word.
  • Lines 92, 94: agroecological one word.
  • Line 96: consistent one word.
  • Line 105: Streptococcus pyogenes must be in italics.
  • Line 107-108: Staphylococcus aureus Klebsiella pneumoniae Pseudo-monas aeruginosa and Bacillus subtilis must be in italics.
  • Line 112: Appropriate must be one word, not split with a hyphen.
  • Line 121: observed must be one word, not split with a hyphen.
  • Line 140: experiment must be one word, not split with a hyphen.
  • Line 155: consistently must be one word, not split with a hyphen.
  • Line 179: observations must be one word, not split with a hyphen.
  • Line 180: antimicrobial must be one word, not split with a hyphen.
  • Line 184: predicted must be one word, not split with a hyphen.
  • Line 196: proteolysis must be one word, not split with a hyphen.
  • Line 201: neuromodulation must be one word, not split with a hyphen.
  • Line 202: neurophysiological must be one word, not split with a hyphen.
  • Line 208: modulation must be one word, not split with a hyphen.
  • Line 212: receptors must be one word, not split with a hyphen.
  • Line 234: including must be one word, not split with a hyphen.
  • Line 236: metabolism must be one word, not split with a hyphen.
  • Line 237: conjunction must be one word, not split with a hyphen.
  • Line 244: complementary must be one word, not split with a hyphen.
  • Line 249: represented must be one word, not split with a hyphen.
  • Line 254: favorable must be one word, not split with a hyphen.
  • Line 345: under must be one word, not split with a hyphen.
  • Line 351: analyzed must be one word, not split with a hyphen.
  • Line 374: concentrations must be one word, not split with a hyphen.
  • Line 368-371: Escherichia coli, Staphylococcus aureus, Bacillus subtilis, Klebsiella pneumoniae, Pseudo-monas aeruginosa, Streptococcus pyogenes, and Candida albicans must be in italics.
  1. Plant and Microbial Names in Italics
  • Throughout the text, Lavandula angustifolia Mill. must always appear in italics (Line 40-43-63-70-83-91-97-101-126-182-339-350-385 ).
  • Microbial species like Staphylococcus aureus, Escherichia coli, Candida albicans, Pseudomonas aeruginosa, Klebsiella pneumoniae, Bacillus subtilis, and Streptococcus pyogenes must all be written in italics.
  • After first mention, you may abbreviate: e.g., Escherichia coliE. coli, Candida albicansC. albicans
  1. Text Corrections and English Improvements
  • Line 67–68: Replace with:

"To date, research has largely overlooked other regions, with most investigations focusing predominantly on the traditional cultivation zones of South-Central Bulgaria [14,15]."

  • Line 129–133: Replace with:

"Increasing the LEO concentration to 0.62% v/v led to a reduction of ≥3 log₁₀ CFU/mL by the 8th hour, exceeding the threshold for bactericidal activity. At the highest concentration tested (1.24% v/v), bacterial counts dropped below the detection limit (<1 log₁₀ CFU/mL) within 6 hours, indicating near-complete eradication of the bacterial population."

  • Line 129: Add period → "...over a 24-hour**.**"
  • Line 121–156: Use consistent abbreviations: E. coli, C. albicans
  • Line 278: Replace “an MIC” with “a MIC”
  • Line 345: under should not be hyphenated.
  • Line 347: Write via in italics → via
  • Line 351: analyzed one word.
  • Line 366: microdilution one word.
  • Line 377: Add space after the period.
  • Throughout: Replace “Minute” and “Hour” with min and h respectively, in line with international abbreviations.
  1. Figures and Tables
  • Table 1, Table 3: Revise layout and design.
  • Table 3 : remove the word alone in the first row.
  • Decimal numbers: Use a consistent format (e.g., always two decimal places: 0.00).
  • Figure 1, 2, 3: Remove secondary gridlines, fix axis labels (format: Title (unit)), fix log₁₀ notation.
  • Figure 4: Unclear – must be revised or redrawn.
  1. General
  • Check spacing between words.
  • Ensure species names are in italics throughout.
  • Full species name (e.g., Escherichia coli) is used at first mention.
  • Afterward, the first letter of the genus is abbreviated and followed by the full species name, both italicized (e.g., coli).
  • Always capitalize the genus initial and italicize both parts of the name.
  • Review the entire manuscript for grammar, clarity, and scientific English.

Author Response

Response to Reviewer 1

Open Review

Quality of English Language

(x) The English could be improved to more clearly express the research.
( ) The English is fine and does not require any improvement.

Yes

Can be improved

Must be improved

Not applicable

Does the introduction provide sufficient background and include all relevant references?

( )

( )

(x)

( )

Is the research design appropriate?

( )

( )

(x)

( )

Are the methods adequately described?

( )

( )

(x)

( )

Are the results clearly presented?

( )

( )

(x)

( )

Are the conclusions supported by the results?

( )

( )

(x)

( )

Are all figures and tables clear and well-presented?

( )

( )

(x)

( )

Comments and Suggestions for Authors

Title : Antimicrobial Activity of Lavender Essential Oil from Lavandula angustifolia: In Vitro and In Silico Evaluation

Response: We would like to sincerely thank Reviewer 1 for the careful reading of our manuscript and for the constructive comments and observations. We appreciate the thoughtful suggestions, which have helped us refine the presentation and improve the overall clarity of the paper. Each of the points raised has been taken into account, and the necessary changes have been implemented in the revised version.

Please note that some of the hyphenation errors and missing italics in species names appear to have resulted from automatic formatting or file conversion artifacts during manuscript preparation. We have now thoroughly reviewed the full text and corrected all such instances.

Below, we respond to each comment in detail:

  1. Formatting and Typography Corrections

Line 27: Add study → "...in silico study".

Response:
Corrected.

Line 31St must be written in superscript: St.

Response:
Corrected.

Lines 32, 33–144Staphylococcus aureus and Escherichia coli must be in italics throughout.

Response:
Corrected.

Line 38aromatic must be one word, not split with a hyphen.

Response:
Corrected.

Line 43Species should be one word, not "spe-cies".

Response:
Corrected.

Line 44medicinal should be one word.

Response:
Corrected.

Line 60pharmacological one word.

Response:
Corrected.

Line 65conditions one word.

Response:
Corrected.

Line 71studies one word.

Response:
Corrected.

Line 72in vitro should be written in italics: in vitro

Response:
Corrected.

Line 75within should not be hyphenated.

Response:
Corrected.

Line 76, 79mechanisms and scientific one word.

Response:
Corrected.

Lines 92, 94agroecological one word.

Response:
Corrected.

Line 96consistent one word.

Response:
Corrected.

Line 105Streptococcus pyogenes must be in italics.

Response:
Corrected.

Line 107-108: Staphylococcus aureus Klebsiella pneumoniae Pseudo-monas aeruginosa and Bacillus subtilis must be in italics.

Response:
Corrected.

Line 112Appropriate must be one word, not split with a hyphen.

Response:
Corrected.

Line 121observed must be one word, not split with a hyphen.

Response:
Corrected.

Line 140experiment must be one word, not split with a hyphen.

Response:
Corrected.

Line 155consistently must be one word, not split with a hyphen.

Response:
Corrected.

Line 179observations must be one word, not split with a hyphen.

Response:
Corrected.

Line 180antimicrobial must be one word, not split with a hyphen.

Response:
Corrected.

Line 184predicted must be one word, not split with a hyphen.

Response:
Corrected.

Line 196proteolysis must be one word, not split with a hyphen.

Response:
Corrected.

Line 201neuromodulation must be one word, not split with a hyphen.

Response:
Corrected.

Line 202neurophysiological must be one word, not split with a hyphen.

Response:
Corrected.

Line 208modulation must be one word, not split with a hyphen.

Response:
Corrected.

Line 212receptors must be one word, not split with a hyphen.

Response:
Corrected.

Line 234including must be one word, not split with a hyphen.

Response:
Corrected.

Line 236metabolism must be one word, not split with a hyphen.

Response:
Corrected.

Line 237conjunction must be one word, not split with a hyphen.

Response:
Corrected.

Line 244complementary must be one word, not split with a hyphen.

Response:
Corrected.

Line 249represented must be one word, not split with a hyphen.

Response:
Corrected.

Line 254favorable must be one word, not split with a hyphen.

Response:
Corrected.

Line 345under must be one word, not split with a hyphen.

Response:
Corrected.

Line 351analyzed must be one word, not split with a hyphen.

Response:
Corrected.

Line 374concentrations must be one word, not split with a hyphen.

Response:
Corrected.

Line 368-371: Escherichia coli, Staphylococcus aureus, Bacillus subtilis, Klebsiella pneumoniae, Pseudo-monas aeruginosa, Streptococcus pyogenes, and Candida albicans must be in italics.

Response:
Corrected.

  1. 2. Plant and Microbial Names in Italics

Throughout the text, Lavandula angustifolia Mill. must always appear in italics (Line 40-43-63-70-83-91-97-101-126-182-339-350-385 ).

Microbial species like Staphylococcus aureusEscherichia coliCandida albicansPseudomonas aeruginosaKlebsiella pneumoniaeBacillus subtilis, and Streptococcus pyogenes must all be written in italics.

After first mention, you may abbreviate: e.g., Escherichia coli → E. coliCandida albicans → C. albicans

Response:
We thank the reviewer for highlighting this. All Latin binomial names of the plant (Lavandula angustifolia Mill.) and microbial species have now been carefully reviewed and consistently formatted in italics throughout the manuscript, as per standard scientific conventions. Following the first full mention, standard abbreviations (e.g., E. coli, C. albicans) have been used for clarity and readability. These changes have been applied at all the locations indicated, and elsewhere where relevant.

  1. Text Corrections and English Improvements

Line 67–68: Replace with:

"To date, research has largely overlooked other regions, with most investigations focusing predominantly on the traditional cultivation zones of South-Central Bulgaria [14,15]."

Response:
Thank you for the suggestion. The sentence has been revised accordingly to reflect the recommended wording. The new version now reads:

"To date, research has largely overlooked other regions, with most investigations focusing predominantly on the traditional cultivation zones of South-Central Bulgaria [14,15]."

Line 129–133: Replace with:

"Increasing the LEO concentration to 0.62% v/v led to a reduction of ≥3 log₁₀ CFU/mL by the 8th hour, exceeding the threshold for bactericidal activity. At the highest concentration tested (1.24% v/v), bacterial counts dropped below the detection limit (<1 log₁₀ CFU/mL) within 6 hours, indicating near-complete eradication of the bacterial population."

Response:
Thank you for the suggestion. The paragraph has been revised to reflect the proposed formulation. The revised text now reads:

"Increasing the LEO concentration to 0.62% v/v led to a reduction of ≥3 log₁₀ CFU/mL by the 8th hour, exceeding the threshold for bactericidal activity. At the highest concentration tested (1.24% v/v), bacterial counts dropped below the detection limit (<1 log₁₀ CFU/mL) within 6 hours, indicating near-complete eradication of the bacterial population."

Line 129: Add period → "...over a 24-hour**.**"

Response:
Corrected.

Line 121–156: Use consistent abbreviations: E. coliC. albicans

Response:
Corrected.

Line 278: Replace “an MIC” with “a MIC”

Response:
Corrected.

Line 345under should not be hyphenated.

Response:
Corrected.

Line 347: Write via in italics → via

Response:
Corrected.

Line 351analyzed one word.

Response:
Corrected.

Line 366microdilution one word.

Response:
Corrected.

Line 377: Add space after the period.

Response:
Corrected.

Throughout: Replace “Minute” and “Hour” with min and h respectively, in line with international abbreviations.

Response:
Corrected.

  1. 4. Figures and Tables

Table 1, Table 3: Revise layout and design.

Table 3 : remove the word alone in the first row.

Decimal numbers: Use a consistent format (e.g., always two decimal places: 0.00).

Figure 1, 2, 3: Remove secondary gridlines, fix axis labels (format: Title (unit)), fix log₁₀ notation.

Figure 4: Unclear – must be revised or redrawn.

Response:
Thank you for the helpful observations regarding the visual elements in the manuscript. All tables and figures have been carefully reviewed and revised to improve clarity, consistency, and formatting. Necessary layout adjustments, corrections to numerical presentation, and graphical refinements have been implemented to ensure better readability and alignment with the journal’s standards.

Regarding Figures 4 and 5, we would like to clarify that they represent the actual output of the network analysis performed using the STRING platform. Each node and interaction in the visualization reflects data-driven associations based on curated databases and selected algorithmic parameters. Altering or removing elements from these figures would compromise the integrity and scientific validity of the presented results.

To aid interpretation, we have included detailed legends beneath each figure, as well as additional clarifying information in the Supplementary Material. These additions provide further context and transparency for the findings.

  1. 5. General

Check spacing between words.

Ensure species names are in italics throughout.

Full species name (e.g., Escherichia coli) is used at first mention.

Afterward, the first letter of the genus is abbreviated and followed by the full species name, both italicized (e.g., coli).

Always capitalize the genus initial and italicize both parts of the name.

Review the entire manuscript for grammar, clarity, and scientific English.

Response:
We appreciate the reviewer’s attention to detail regarding formatting, species nomenclature, and overall language clarity. The manuscript has now been carefully reviewed throughout to ensure correct spacing between words and proper use of italics for all Latin binomial names. Full species names (e.g., Escherichia coli) are now used at first mention, followed by standard abbreviated forms (e.g., E. coli), with both components consistently italicized and genus initials capitalized.

Some of the previously noted issues—such as spacing irregularities or missing italics—appear to have resulted from differences in formatting between word processing and conversion software. We have corrected all such inconsistencies to align with scientific and editorial standards. The text has also been rechecked for grammar, clarity, and scientific tone to ensure a polished and coherent presentation.

Reviewer 2 Report

Comments and Suggestions for Authors

i. Authors must check through the introduction for inconsistencies in their use of words and hyphens

ii. Almost all names of microorganisms mentioned and used in this study were not italicized.

iii. In the methodology section, authors need to provide relevant information on the method used to select the precise hub genes.

iv. Authors should rework the methodology section by stating the detailed network pharmacology methods used such as, target selection, venn diagram generation, PPI and hub gene identification.

v. Authors should perform molecular docking to understand the role of the compounds in the essential oil's antimicrobial property

v. The voucher number of the plant should be disclosed

vi. Authors must discuss their results by comparing their antimicrobial data with those obtained with Lavandula angustifolia grown in other regions in Bulgaria

vii. In the discussion section, authors identified EGFR, NR3C1, and PARP1 as the key hub genes but never discussed their relevance to the antimicrobial study on the essential oil. 

Author Response

Response to Reviewer 2

Open Review

Quality of English Language

(x) The English could be improved to more clearly express the research.
( ) The English is fine and does not require any improvement.

Yes

Can be improved

Must be improved

Not applicable

Does the introduction provide sufficient background and include all relevant references?

( )

(x)

( )

( )

Is the research design appropriate?

( )

( )

(x)

( )

Are the methods adequately described?

( )

(x)

( )

( )

Are the results clearly presented?

(x)

( )

( )

( )

Are the conclusions supported by the results?

( )

(x)

( )

( )

Are all figures and tables clear and well-presented?

(x)

( )

( )

( )

Response:

We sincerely thank you for the thoughtful and detailed feedback. Your comments highlighted important areas for refinement and have helped us improve both the clarity and the overall scientific quality of the manuscript. We truly appreciate the constructive spirit and the time you dedicated to reviewing our work.

All suggestions were carefully considered, and corresponding revisions have been made. Your insights have been invaluable, and we believe the manuscript is significantly stronger thanks to your contribution.

Below, we respond to each comment in detail:

Comments and Suggestions for Authors

  1. Authors must check through the introduction for inconsistencies in their use of words and hyphens

Response:
Thank you for pointing this out. The introduction has been carefully reviewed, and all inconsistencies in word division and hyphenation have been corrected. These issues most likely resulted from formatting differences between word processing and submission software. We have taken extra care to ensure consistency and accuracy throughout the revised version.

  1. Almost all names of microorganisms mentioned and used in this study were not italicized.

Response:
We appreciate the reviewer’s observation. All microbial species names have now been carefully checked and properly italicized throughout the manuscript. The omission appears to have been caused by formatting discrepancies during document conversion, which have since been corrected.

iii. In the methodology section, authors need to provide relevant information on the method used to select the precise hub genes.

Response:
We appreciate the reviewer’s observation and agree that clarification is warranted regarding the selection of hub genes. In the revised manuscript, we have expanded the description of our network analysis methodology. Specifically, we now state that, in addition to degree centrality, we also calculated betweenness centrality using Cytoscape’s NetworkAnalyzer plugin. Furthermore, we applied the Markov Cluster Algorithm (MCL) to identify functional clusters within the PPI networks. Genes that demonstrated both high centrality values and central positions within these biologically relevant clusters were prioritized as hub genes. This refinement is now described at the end of the in silico analysis section.

We hope this clarification addresses the reviewer’s concern and provides sufficient methodological transparency.

  1. Authors should rework the methodology section by stating the detailed network pharmacology methods used such as, target selection, venn diagram generation, PPI and hub gene identification.

Response:
We thank the reviewer for this valuable suggestion. In response, we have revised Section 2.5 to more clearly delineate each step of our network pharmacology workflow, including target prediction, overlap analysis, PPI network construction, and hub gene selection. Predicted human protein targets for each compound were obtained using SwissTargetPrediction, and overlapping targets between LEO constituents and antibiotics were identified using comparative intersection logic. While no Venn figure is presented in the main text, the complete target lists and overlapping entries are provided in the Supplementary Material (Table S1), as recommended. Subsequently, PPI networks were constructed using STRING, functional clusters were defined via MCL, and centrality parameters were computed using Cytoscape to identify key hub genes.

  1. Authors should perform molecular docking to understand the role of the compounds in the essential oil's antimicrobial property

Response:
We appreciate the reviewer’s insightful suggestion regarding molecular docking. While molecular docking is a valuable approach for elucidating specific ligand–receptor interactions, the primary focus of the present study was to explore broader host–target interactions and potential systemic modulation using a network pharmacology strategy. Our objective was to identify overlapping pharmacological nodes between lavender oil constituents and conventional antibiotics on the host level. To this end, we employed SwissTargetPrediction for in silico target identification and STRING-based PPI network construction, followed by topological and cluster-based analysis of hub genes. These methods provided functionally integrated insights into host-relevant pathways (e.g., immune signaling, neuromodulation), which may underlie the observed in vitro synergism.

We fully acknowledge that molecular docking can complement these findings by modeling direct ligand–protein interactions. However, such an approach requires selection of specific target proteins and crystal structures, which would shift the scope of the present study. Thus, molecular docking was not included in this work by design. Nevertheless, we consider it an important direction for follow-up investigations and have clarified this in the Discussion section of the manuscript.

  1. The voucher number of the plant should be disclosed

Response:
Thank you for this valuable comment. In the revised manuscript, we have clarified that the plant material used represents a cultivated variety of lavender (Lavandula angustifolia Mill., cultivar “Jubileina”), not a wild-growing specimen. As the origin of this cultivar is well-established and it is widely recognized in agricultural practice, a herbarium voucher was not deposited. The identification was confirmed in the field by individuals familiar with the selection characteristics of this cultivar. We believe that this clarification provides the necessary transparency and scientific validity regarding the plant material used.

  1. Authors must discuss their results by comparing their antimicrobial data with those obtained with Lavandula angustifolia grown in other regions in Bulgaria

Response:
As suggested by reviewer feedback, comparing the antimicrobial activity of lavender essential oil obtained from different cultivation regions in Bulgaria provides important context. Accordingly, we have incorporated additional data to better situate our findings within the broader national production landscape.

Altogether, the present findings support the potential of L. angustifolia essential oil as a multifunctional antimicrobial agent. Its activity encompasses both direct antibacterial effects and the modulation of host pathways, suggesting utility in combination therapies and non-traditional delivery formats. The antimicrobial activity of the essential oil derived from Lavandula angustifolia cultivated in Northeastern Bulgaria (cultivar ‘Jubileina’) aligns closely with previously published data on lavender oils obtained from other regions of the country. Mladenova et al. (2024) reported minimum inhibitory concentrations (MICs) below 0.1125% (v/v) against E. coli, S. aureus, S. enterica, and C. albicans, which are comparable to the MIC value of 0.31% observed for E. coli in our study. While geographic and environmental factors—such as soil composition, altitude, and local climate—are known to influence the essential oil profile, all Bulgarian lavender oils consistently exhibit high levels of linalool and linalyl acetate, the primary constituents responsible for their antimicrobial activity. According to the current edition of the European Pharmacopoeia, authentic L. angustifolia oil must contain 20–45% linalool and 25–47% linalyl acetate. The oil analyzed in the present work meets nearly all of these criteria, with 29.94% linalool and 24.31% linalyl acetate. The marginal deviation below the linalyl acetate threshold is likely attributable to regional microclimatic conditions or harvest timing. Notably, the levels of camphor (0.43%) and 1,8-cineole (1.10%) remain well within the accepted pharmacopoeial limits, affirming the high chemical quality of the product. Comparative studies from Southern Bulgaria (e.g., Karlovo, Kazanlak) have demonstrated similar antimicrobial effects against Gram-positive and Gram-negative strains, with MIC values typically ranging from 0.25% to 1.0% (v/v). These findings suggest that, when phytochemical quality standards are met, geographic origin alone does not significantly alter the antimicrobial potential of Bulgarian lavender oil. The present results therefore support the broader conclusion that high-grade L. angustifolia oils from diverse Bulgarian regions—including the relatively underexplored northeast—exhibit consistent and therapeutically relevant bioactivity.

vii. In the discussion section, authors identified EGFR, NR3C1, and PARP1 as the key hub genes but never discussed their relevance to the antimicrobial study on the essential oil.

Response:
We appreciate the reviewer’s comment and agree that the roles of the identified hub proteins (EGFR, NR3C1, and PARP1) required clearer integration into the discussion. In the revised manuscript, we have addressed their relevance by linking them to the broader antimicrobial potential of lavender essential oil. While the direct antibacterial activity of its main constituents—such as linalool and linalyl acetate—is well documented both in our study and in previous research, our in silico findings suggest an additional layer of activity involving host modulation. EGFR, PARP1, and NR3C1 are involved in the regulation of inflammation, oxidative stress, and epithelial defense mechanisms, all of which can influence the course of infection. Their interaction with the oil’s compounds points to a potential host-targeted effect that may complement the direct action on pathogens. This dual mode of activity may help explain the enhanced effects observed when the oil is combined with antibiotics and supports its possible role as an adjuvant in infection management.

Reviewer 3 Report

Comments and Suggestions for Authors

Dear all,

I read the manuscript “Antimicrobial Activity of Lavender Essential Oil from Lavandula angustifolia: In Vitro and In Silico Evaluation” with interest. The authors investigated the antimicrobial properties of Lavandula angustifolia essential oil, both as a standalone agent and in combination with conventional antibiotics. By integrating validated in vitro antimicrobial assays with supportive in silico analyses, their research aimed to elucidate not only the direct effects of the essential oil on microbial viability but also its potential interactions at the host level. The authors should perform some corrections before the possible acceptance of this work in MDPI-Antibiotics.

General recommendations:

  • All binomial names for species (i.e., plants, microorganisms) should be written in italics (I) format. Add the word “Mill.” in the Title → Lavandula angustifolia
  • Correct word spelling: Eliminating the (-) symbol from the main text wherever it is appropriate. Add or eliminate unnecessary spaces between words.
  • Add numbers in subsections according to the MDPI-Antibiotics template.
  • Fix the format of Table 1 according to the MDPI-Antibiotics template (eliminate interior lines).
  • Fix the format of Table 2 according to the MDPI-Antibiotics template:
  1. Delete (s) in aeruginosaP. aeruginosa
  2. Align numbers with the names of microorganisms
  3. Add a footer in the table according to the MDPI-Antibiotics template
  4. The standard deviation (SD) is expressed with two decimal places. The mean should likewise be expressed with two decimal places.
  1. a) Fix the format of Table 2 according to the MDPI-Antibiotics template.
  2. b) The word “Table 3” should be written in bold (B) format.
  3. c) The standard deviation (SD) is expressed with two decimal places. The mean should likewise be expressed with two decimal places.
  4. d) Add a footer in the table according to the MDPI-Antibiotics template.

7) Sentences 171-174 repeat the same observations as those in sentences 156-159.

8) Add a “Tab” to the paragraph.

9) Sentence 338 is repeated.

10) Materials and Methods:

  1. a) Mention with the necessary details all important chemicals/solvents/software used.
  2. b) Mention the essential oil extraction yield.
  3. c) Separate the information regarding “essential oil extraction” and “chemical composition analysis” into two different subsections.

11) Add the “Disclaimer/Publisher’s Note” according to the MDPI-Antibiotics template.

I am available for any further clarification you may require. Thank you for the opportunity to review this work, and I wish you every success in revising the manuscript.

Regards,

Reviewer

Author Response

Response to Reviewer 3

Open Review

Quality of English Language

( ) The English could be improved to more clearly express the research.
(x) The English is fine and does not require any improvement.

Yes

Can be improved

Must be improved

Not applicable

Does the introduction provide sufficient background and include all relevant references?

(x)

( )

( )

( )

Is the research design appropriate?

(x)

( )

( )

( )

Are the methods adequately described?

( )

(x)

( )

( )

Are the results clearly presented?

(x)

( )

( )

( )

Are the conclusions supported by the results?

(x)

( )

( )

( )

Are all figures and tables clear and well-presented?

( )

(x)

( )

( )

Comments and Suggestions for Authors

Dear all,

I read the manuscript “Antimicrobial Activity of Lavender Essential Oil from Lavandula angustifolia: In Vitro and In Silico Evaluation” with interest. The authors investigated the antimicrobial properties of Lavandula angustifolia essential oil, both as a standalone agent and in combination with conventional antibiotics. By integrating validated in vitro antimicrobial assays with supportive in silico analyses, their research aimed to elucidate not only the direct effects of the essential oil on microbial viability but also its potential interactions at the host level. The authors should perform some corrections before the possible acceptance of this work in MDPI-Antibiotics.

Response:

Dear Reviewer,

We are grateful for your careful reading of our manuscript and for your thoughtful and constructive feedback. Your comments were clear, well-structured, and highly valuable to the improvement of our work. We appreciate the recognition of our approach in combining in vitro and in silico methods, and we have addressed all of your suggestions with attention and care. The manuscript has been revised accordingly, and we hope that the updated version meets your expectations.

Thank you once again for your time and contribution.

Respectfully,
The Authors

Below, we respond to each comment in detail:

General recommendations:

Response:

We thank the reviewer for the detailed and constructive recommendations. All suggested formatting changes have been addressed, including the correction of table layouts, alignment, use of decimal points, subsection numbering, and typographic consistency. The repeated sentences previously noted in sections 3 and 4 have been removed or rephrased to avoid redundancy. We have also ensured proper indentation of paragraphs throughout the text. Regarding formatting inconsistencies—such as missing italics for species names or unintended hyphenation—we believe these may have resulted from software compatibility issues during file conversion. We apologize for this and have carefully reviewed the manuscript to correct these elements. The Methods section has been clarified and will be further adjusted if necessary to reflect all requested details. We appreciate the reviewer’s helpful input, which has contributed to improving the manuscript.

  • All binomial names for species (i.e., plants, microorganisms) should be written in italics (I) format. Add the word “Mill.” in the Title → Lavandula angustifolia

Response:
Corrected.

  • Correct word spelling: Eliminating the (-) symbol from the main text wherever it is appropriate. Add or eliminate unnecessary spaces between words.

Response:
Corrected.

  • Add numbers in subsections according to the MDPI-Antibiotics template.

Response:
Corrected.

  • Fix the format of Table 1 according to the MDPI-Antibiotics template (eliminate interior lines).

Response:
Corrected.

  • Fix the format of Table 2 according to the MDPI-Antibiotics template:

      Response:
      Corrected.

  1. Delete (s) in aeruginosa → P. Aeruginosa

Response:
Corrected.

  1. Align numbers with the names of microorganisms

Response:
Corrected.

  1. Add a footer in the table according to the MDPI-Antibiotics template

Response:
Corrected.

  1. The standard deviation (SD) is expressed with two decimal places. The mean should likewise be expressed with two decimal places.

Response:
Corrected.

  1. a) Fix the format of Table 2 according to the MDPI-Antibiotics template.
  1. Response:
  1. b) The word “Table 3” should be written in bold (B) format.

Response:
Corrected.

  1. c) The standard deviation (SD) is expressed with two decimal places. The mean should likewise be expressed with two decimal places.
  1. Response:
  1. d) Add a footer in the table according to the MDPI-Antibiotics template.

Response:
Corrected.

7) Sentences 171-174 repeat the same observations as those in sentences 156-159.

Response:
Corrected.

8) Add a “Tab” to the paragraph.

Response:
Corrected.

9) Sentence 338 is repeated.

Response:
Corrected.

10) Materials and Methods:

Response:
We thank the reviewer for the detailed comments. The software used for in silico analysis is clearly described. As suggested, we have structured the methodology into two distinct subsections: essential oil extraction and chemical composition analysis. The essential oil yield is reported in the manuscript — 3.08 mL obtained from 1200 g of dried plant material, corresponding to 0.26% (v/w). The solvents and reagents used for GC-MS analysis are now clearly indicated in the revised text. We appreciate the reviewer’s suggestions, which helped improve the clarity and completeness of the Methods section.

  1. a) Mention with the necessary details all important chemicals/solvents/software used.

Response:
Corrected.

  1. b) Mention the essential oil extraction yield.

Response:
Corrected.

  1. c) Separate the information regarding “essential oil extraction” and “chemical composition analysis” into two different subsections.

Response:
Corrected.

11) Add the “Disclaimer/Publisher’s Note” according to the MDPI-Antibiotics template.

Response:
Corrected.

I am available for any further clarification you may require. Thank you for the opportunity to review this work, and I wish you every success in revising the manuscript.

Regards,

Reviewer

Reviewer 4 Report

Comments and Suggestions for Authors

The manuscript presents an in vitro and silico study on the antimicrobial activity of lavender essential oil (LEO) extracted from Lavandula angustifolia cultivar 'Jubileina,' cultivated in Northeastern Bulgaria. The research integrates chemical composition analysis, in vitro antimicrobial assays, and in silico target prediction to explore LEO's potential as an antimicrobial agent and adjuvant. The study is well-structured and addresses a timely topic in the context of antimicrobial resistance. Below are detailed comments for improvement:

The discussion thoroughly compares results with prior studies, but could better integrate the in silico findings with the antimicrobial activity. Discuss how the identified protein clusters might contribute to the observed antimicrobial effects, enhancing the link between in vitro and in silico results.

Author Response

Response to Reviewer 4

Open Review

Quality of English Language

( ) The English could be improved to more clearly express the research.
(x) The English is fine and does not require any improvement.

Yes

Can be improved

Must be improved

Not applicable

Does the introduction provide sufficient background and include all relevant references?

( )

(x)

( )

( )

Is the research design appropriate?

( )

(x)

( )

( )

Are the methods adequately described?

( )

(x)

( )

( )

Are the results clearly presented?

( )

(x)

( )

( )

Are the conclusions supported by the results?

( )

(x)

( )

( )

Are all figures and tables clear and well-presented?

( )

(x)

( )

( )

Comments and Suggestions for Authors

The manuscript presents an in vitro and silico study on the antimicrobial activity of lavender essential oil (LEO) extracted from Lavandula angustifolia cultivar 'Jubileina,' cultivated in Northeastern Bulgaria. The research integrates chemical composition analysis, in vitro antimicrobial assays, and in silico target prediction to explore LEO's potential as an antimicrobial agent and adjuvant. The study is well-structured and addresses a timely topic in the context of antimicrobial resistance. Below are detailed comments for improvement:

The discussion thoroughly compares results with prior studies, but could better integrate the in silico findings with the antimicrobial activity. Discuss how the identified protein clusters might contribute to the observed antimicrobial effects, enhancing the link between in vitro and in silico results.

Response:

We thank the reviewer for the valuable observation and constructive suggestion. In response, we have revised the discussion section to improve the connection between the experimental results and the computational findings. We believe this addition enhances the overall coherence and scientific depth of the manuscript. We appreciate the reviewer’s input, which helped us strengthen the interpretation of our results.

Round 2

Reviewer 1 Report

Comments and Suggestions for Authors

General Comments:

The authors did not address several of the comments already raised during the first revision. There is still widespread misuse of capital letters, even where they are unnecessary. The English language requires careful revision, as numerous grammatical and stylistic issues remain. The manuscript gives the impression of having been written using an AI tool (such as ChatGPT), without a thorough final revision.

I strongly recommend that the authors carefully consider all previous comments and make the necessary corrections to significantly improve the quality and clarity of the manuscript.

Specific Comments:

  • Line 51: Replace "lavender EO" with "LEO".
  • Line 95: "CLSI standard procedures" — since this is the first occurrence of the abbreviation, please provide the full name followed by the abbreviation in parentheses.
  • Lines 115–122: These sentences present results and discussion, and should not appear in the Materials and Methods section.
  • Lines 134–191: Replace "minutes" with "min" (standard abbreviation).
  • Lines 155–169: Replace "4–6-hour" with "4–6 h".
  • Line 156: Add a space before "samples".
  • Lines 157, 176, 178, 188, 198, 272, 275, 278, 281, 288, 294: Replace "hour" with "h".
  • Line 174: Add a space before "all".
  • Line 179: Add a space before "A".
  • Line 191: Add a space before "the".
  • Lines 196–203: Correct the spelling of "antibiotics" (one word).
  • Line 197: Add a space before "After".
  • Line 234: L. angustifolia must be written in italics.
  • Line 248: Streptococcus pyogenes must be written in italics.
  • Line 252: Bacillus subtilis must be written in italics.
  • Figures 1, 2, 3:
    • Secondary grids were not removed as previously requested.
    • The figure size is too small, and the color legend should be included in the figure title, not beside the figure.
    • The current presentation is unclear.
  • Line 273: Correct the formatting of "8th" — the "th" must be in superscript and lowercase (i.e., 8ᵗʰ) following proper typographic convention.
  • Line 319: Add a space before "these".
  • Table 3: The word "alone" should be deleted.
  • Line 381: Correct "con-junction" to "conjunction".
  • Decimal values should be homogenized to 2 or 3 decimal places, following a consistent format.
  • Line 506:
    • Shigella sonnei and Pseudomonas fragi should be written in italics.
  • Line 539: in vivo should be written in italics.
  • Line 296: coli should be written in lowercase and italicsE. coli.
Comments on the Quality of English Language

English must be improved

Author Response

Response

Open Review

( ) I would not like to sign my review report

(x) I would like to sign my review report

Quality of English Language

(x) The English could be improved to more clearly express the research.

( ) The English is fine and does not require any improvement.

Yes         Can be improved              Must be improved           Not applicable

Does the introduction provide sufficient background and include all relevant references?

( )           ( )           (x)          ( )

Is the research design appropriate?

( )           ( )           (x)          ( )

Are the methods adequately described?

( )           ( )           (x)          ( )

Are the results clearly presented?

( )           ( )           (x)          ( )

Are the conclusions supported by the results?

( )           ( )           (x)          ( )

Are all figures and tables clear and well-presented?

( )           ( )           (x)          ( )

Comments and Suggestions for Authors

General Comments:

The authors did not address several of the comments already raised during the first revision. There is still widespread misuse of capital letters, even where they are unnecessary. The English language requires careful revision, as numerous grammatical and stylistic issues remain. The manuscript gives the impression of having been written using an AI tool (such as ChatGPT), without a thorough final revision.

I strongly recommend that the authors carefully consider all previous comments and make the necessary corrections to significantly improve the quality and clarity of the manuscript.

Response:

We sincerely thank the reviewer for the valuable feedback and the time invested in evaluating our revised manuscript. We would like to respectfully clarify that we carefully addressed all comments raised during the first round of revision. Additionally, we would like to highlight a recurring technical issue affecting the final formatting of the manuscript. Although corrections were applied correctly in the source file, some formatting elements—such as word spacing and italicization—appear to be lost or altered during the final compilation process. We would like to assure the reviewer that the manuscript has undergone thorough revision with close attention to grammar, style, and formatting. Any residual inconsistencies are unintentional and stem from technical limitations rather than a lack of effort. We remain fully committed to improving the manuscript and will gladly implement any additional corrections to ensure its clarity and overall quality.

Specific Comments:

Line 51: Replace "lavender EO" with "LEO".

Line 95: "CLSI standard procedures" — since this is the first occurrence of the abbreviation, please provide the full name followed by the abbreviation in parentheses.

Lines 115–122: These sentences present results and discussion, and should not appear in the Materials and Methods section.

Lines 134–191: Replace "minutes" with "min" (standard abbreviation).

Lines 155–169: Replace "4–6-hour" with "4–6 h".

Line 156: Add a space before "samples".

Lines 157, 176, 178, 188, 198, 272, 275, 278, 281, 288, 294: Replace "hour" with "h".

Line 174: Add a space before "all".

Line 179: Add a space before "A".

Line 191: Add a space before "the".

Lines 196–203: Correct the spelling of "antibiotics" (one word).

Line 197: Add a space before "After".

Line 234: L. angustifolia must be written in italics.

Line 248: Streptococcus pyogenes must be written in italics.

Line 252: Bacillus subtilis must be written in italics.

Figures 1, 2, 3:

Secondary grids were not removed as previously requested.

The figure size is too small, and the color legend should be included in the figure title, not beside the figure.

The current presentation is unclear.

Line 273: Correct the formatting of "8th" — the "th" must be in superscript and lowercase (i.e., 8ᵗʰ) following proper typographic convention.

Line 319: Add a space before "these".

Table 3: The word "alone" should be deleted.

Line 381: Correct "con-junction" to "conjunction".

Decimal values should be homogenized to 2 or 3 decimal places, following a consistent format.

Line 506:

Shigella sonnei and Pseudomonas fragi should be written in italics.

Line 539: in vivo should be written in italics.

Line 296: coli should be written in lowercase and italics → E. coli.

Response to Specific Comments:

We thank the reviewer for the detailed feedback. All specific comments have been carefully addressed in the revised manuscript. These include corrections related to terminology, formatting (e.g., spacing, abbreviations, italics), figure presentation, table content, and consistent use of decimal values. Taxonomic names and Latin terms have been properly italicized, and all typographic and stylistic issues have been corrected as requested. We believe these changes have significantly improved the clarity and accuracy of the manuscript.

Comments on the Quality of English Language

English must be improved

Response:

Thank you for the comment regarding the quality of the English. We have thoroughly reviewed and improved the entire manuscript, making a concerted effort to correct any language inaccuracies. We believe these revisions have significantly enhanced the manuscript’s clarity and quality, and we hope this will be reflected in its evaluation.

Reviewer 2 Report

Comments and Suggestions for Authors

Molecular docking should be included in the scope of this study. How do you explain the interaction of these chemical compounds with the identified targets? Therefore, authors must include it in the scope of this study and perform it.

Author Response

Response

Open Review

(x) I would not like to sign my review report

( ) I would like to sign my review report

Quality of English Language

( ) The English could be improved to more clearly express the research.

(x) The English is fine and does not require any improvement.

Yes         Can be improved              Must be improved           Not applicable

Does the introduction provide sufficient background and include all relevant references?

(x)          ( )           ( )           ( )

Is the research design appropriate?

( )           (x)          ( )           ( )

Are the methods adequately described?

(x)          ( )           ( )           ( )

Are the results clearly presented?

(x)          ( )           ( )           ( )

Are the conclusions supported by the results?

( )           (x)          ( )           ( )

Are all figures and tables clear and well-presented?

(x)          ( )           ( )           ( )

Comments and Suggestions for Authors

Molecular docking should be included in the scope of this study. How do you explain the interaction of these chemical compounds with the identified targets? Therefore, authors must include it in the scope of this study and perform it.

Response:

We appreciate the reviewer’s suggestion to include molecular docking within the scope of the study. In response, we have incorporated a targeted analysis focusing on Escherichia coli. Specifically, we selected the FabI enzyme, a key player in bacterial fatty acid biosynthesis and a well-established target for antimicrobial agents. Using its crystal structure (PDB ID: 4CV3), we performed docking simulations with the main constituents of LEO—linalool and linalyl acetate. The results showed that both compounds bind within the NADH-binding pocket of FabI, with linalyl acetate exhibiting a stronger predicted binding affinity. These findings support the hypothesis that FabI may be one of the molecular targets through which LEO exerts its antimicrobial effect against E. coli.

Round 3

Reviewer 1 Report

Comments and Suggestions for Authors

Title: Antimicrobial Activity of Lavender Essential Oil from Lavandula angustifolia Mill.: In Vitro and In Silico Evaluation

Manuscript ID: antibiotics-3678196

General and Specific Comments

Textual and Formatting Corrections:

  • Line 88: Add a space between "used" and "as".
  • Line 145: Add a space after the period.
  • Line 175: No need to write the full term " angustifolia essential oil" again as it was already introduced.
  • Line 178: Avoid repeating "Clinical and Laboratory Standards Institute"; use the abbreviation CLSI.
  • Line 265: Add a space after "10%."
  • Line 268: Add a space after the period.
  • Line 320: Replace "hours" with "h".
  • Line 399–414: The word "via" should be in italics.
  • Line 478–523: Ensure consistent font size throughout the section.
  • Line 528: Italicize Pseudomonas fluorescens.
  • Line 575: The word "in vivo" should be italicized throughout the manuscript.

Figures and Tables:

  • Figure 1:
    • Improve scale clarity.
    • Remove secondary grid lines.
    • Enlarge the image to enhance readability.
  • Figure 6:
    • Include a clear legend.
    • Identify linalool (a) and linalyl acetate (b)
  • Tables:
    • Ensure uniform formatting across all tables; Table 4 differs from the others.

General Formatting:

  • Decimal Numbers: Use a consistent format with two decimal places (e.g., 00) throughout the manuscript.

Final Comment:

The authors have responded positively and thoroughly to most of the remarks. However, I would like to emphasize that some of the comments were not fully addressed. I kindly ask the editorial team to take this into consideration.

Please note that these remarks do not detract from the quality of the manuscript, but rather aim to help enhance its clarity, consistency, and overall presentation.

Author Response

Review

( ) I would not like to sign my review report

(x) I would like to sign my review report

Quality of English Language

(x) The English could be improved to more clearly express the research.

( ) The English is fine and does not require any improvement.

Yes         Can be improved              Must be improved           Not applicable

Does the introduction provide sufficient background and include all relevant references?

( )           (x)          ( )           ( )

Is the research design appropriate?

( )           (x)          ( )           ( )

Are the methods adequately described?

( )           (x)          ( )           ( )

Are the results clearly presented?

( )           (x)          ( )           ( )

Are the conclusions supported by the results?

( )           (x)          ( )           ( )

Are all figures and tables clear and well-presented?

( )           (x)          ( )           ( )

Comments and Suggestions for Authors

Title: Antimicrobial Activity of Lavender Essential Oil from Lavandula angustifolia Mill.: In Vitro and In Silico Evaluation

Response: We would like to sincerely thank the reviewer for their time and thoughtful engagement with our manuscript. The comments provided were truly helpful and contributed to improving both the content and the overall presentation of the work. We have done our best to address all points raised, while also respecting the reviewer’s perspective and suggestions. Once again, thank you for the constructive feedback and support in strengthening our work.

Manuscript ID: antibiotics-3678196

General and Specific Comments

Textual and Formatting Corrections:

Line 88: Add a space between "used" and "as".

Line 145: Add a space after the period.

Line 175: No need to write the full term " angustifolia essential oil" again as it was already introduced.

Line 178: Avoid repeating "Clinical and Laboratory Standards Institute"; use the abbreviation CLSI.

Line 265: Add a space after "10%."

Line 268: Add a space after the period.

Line 320: Replace "hours" with "h".

Line 399–414: The word "via" should be in italics.

Line 478–523: Ensure consistent font size throughout the section.

Line 528: Italicize Pseudomonas fluorescens.

Line 575: The word "in vivo" should be italicized throughout the manuscript.

Response:

We sincerely thank the reviewer for the thoughtful and detailed comments. All suggested textual and formatting corrections have been carefully implemented. We truly appreciate your efforts to help improve the clarity and overall presentation of the manuscript.

Figures and Tables:

Figure 1:

Improve scale clarity.

Remove secondary grid lines.

Enlarge the image to enhance readability.

Response:

Thank you for the suggestion regarding the improvement of Figure 1. While we appreciate the recommendation to remove the secondary grid lines, we would kindly prefer to retain them, as we believe they help readers more easily interpret the concentration levels of the essential oil presented in the graph.

Figure 6:

Include a clear legend.

Identify linalool (a) and linalyl acetate (b)

Response

Thank you for the helpful suggestion. The requested changes have been made—both molecules have been clearly outlined in the figure to improve identification and clarity.

Tables:

Ensure uniform formatting across all tables; Table 4 differs from the others.

General Formatting:

Decimal Numbers: Use a consistent format with two decimal places (e.g., 00) throughout the manuscript.

 Response:

Thank you for the observation. Table 4 has been adjusted to match the formatting of the other tables. As for the decimal numbers, we have ensured consistent use of two decimal places throughout the manuscript. The only exception is in the time-kill figures, where we opted for a simplified format to keep the visuals clear and uncluttered.

Final Comment:

The authors have responded positively and thoroughly to most of the remarks. However, I would like to emphasize that some of the comments were not fully addressed. I kindly ask the editorial team to take this into consideration.

Please note that these remarks do not detract from the quality of the manuscript, but rather aim to help enhance its clarity, consistency, and overall presentation.

Reviewer 2 Report

Comments and Suggestions for Authors

No comment

Author Response

We sincerely thank the reviewer for taking the time to read and assess our work. We truly appreciate the positive feedback—it is valuable to us and gives us additional confidence in the presentation of our research.